# Identification of New Candidate Genes Related to Semen Traits in Duroc Pigs through Weighted Single-Step GWAS

**DOI:** 10.3390/ani13030365

**Published:** 2023-01-20

**Authors:** Xiaoke Zhang, Qing Lin, Weili Liao, Wenjing Zhang, Tingting Li, Jiaqi Li, Zhe Zhang, Xiang Huang, Hao Zhang

**Affiliations:** 1National Engineering Research Center for Breeding Swine Industry, Guangdong Provincial Key Lab of Agro-Animal Genomics and Molecular Breeding, College of Animal Science, South China Agricultural University, Guangzhou 510642, China; 2Guangdong Guyue Technology Co., Ltd. Guangzhou 510980, China

**Keywords:** pig, semen traits, weighted single-step GWAS, candidate genes, gene network

## Abstract

**Simple Summary:**

Due to the complexity of sperm cell reproduction and maturation, the genetic structure of semen traits remains largely unknown. In our study, we used weighted single-step GWAS to detect genetic regions and further candidate genes related to semen traits in Duroc boars. This study provides in-depth understanding of the genetic structure of semen traits and the biological information provided by gene networks, and can be applied to speed up the genetic process of semen traits in boars. The candidate genes *CATSPER1*, *STRA8*, *ZSWIM7*, *TEKT3*, *UBB*, *PTBP2*, *EIF2B2*, *MLH3*, and *CCDC70* were associated with semen traits in Duroc pigs.

**Abstract:**

Semen traits play a key role in the pig industry because boar semen is widely used in purebred and crossbred pigs. The production of high-quality semen is crucial to ensuring a good result in artificial insemination. With the wide application of artificial insemination in the pig industry, more and more attention has been paid to the improvement of semen traits by genetic selection. The purpose of this study was to identify the genetic regions and candidate genes associated with semen traits of Duroc boars. We used weighted single-step GWAS to identify candidate genes associated with sperm motility, sperm progressive motility, sperm abnormality rate and total sperm count in Duroc pigs. In Duroc pigs, the three most important windows for sperm motility—sperm progressive motility, sperm abnormality rate, and total sperm count—explained 12.45%, 9.77%, 15.80%, and 12.15% of the genetic variance, respectively. Some genes that are reported to be associated with spermatogenesis, testicular function and male fertility in mammals have been detected previously. The candidate genes *CATSPER1*, *STRA8*, *ZSWIM7*, *TEKT3*, *UBB*, *PTBP2*, *EIF2B2*, *MLH3*, and *CCDC70* were associated with semen traits in Duroc pigs. We found a common candidate gene, *STRA8*, in sperm motility and sperm progressive motility, and common candidate genes *ZSWIM7*, *TEKT3* and *UBB* in sperm motility and sperm abnormality rate, which confirms the hypothesis of gene pleiotropy. Gene network enrichment analysis showed that *STRA8*, *UBB* and *CATSPER1* were enriched in the common biological process and participated in male meiosis and spermatogenesis. The SNPs of candidate genes can be given more weight in genome selection to improve the ability of genome prediction. This study provides further insight into the understanding the genetic structure of semen traits in Duroc boars.

## 1. Introduction

Semen traits play a key role in the wide application of artificial insemination in the pig industry. The economic profit of artificial pig breeding stations is highly dependent on the quantity and quality of semen [1], and the decline of semen quality is one of the main reasons for shortening the life span of boars [2]. Understanding the genetic background and detecting the genetic markers related to semen traits are helpful to improve the genetic selection of semen traits and speed up the genetic process.

In recent years, with the rapid development of high-throughput genotyping and molecular technology, researchers can accurately identify quantitative trait loci by looking for the association between genetic markers and phenotypic records, which is called genome-wide association study (GWAS) [3]. GWAS has been successfully applied to QTL mapping of important economic traits in animal and plant breeding and detection of genetic risk factors of human diseases [4]. Candidate genes related to semen traits were also identified by GWAS in several previous studies [5,6,7,8]. However, due to the complexity of sperm cell reproduction and maturation, the genetic structure of semen characteristics is still unknown to a large extent [5]. The candidate genes found may be different for the same trait in the same breed of different populations. Gene mutations and impaired expression that control the whole process of spermatogenesis and sperm maturation will lead to problems in semen quality and fertility. Therefore, mining more QTL regions and candidate genes related to semen traits can broaden our understanding of the genetic structure of porcine semen traits.

Weighted single-step GWAS (WssGWAS), proposed by Wang et al. [9], is a method to estimate SNP effect based on the single-step best linear unbiased prediction (ssGBLUP) [10] of genome breeding values (GEBV) of all phenotypic, genotypic and pedigree-related animals. In addition, it allows the variance of SNP to vary, thus improving the accuracy of SNP effect estimation. Therefore, when the number of animals with both phenotype and genotype is small and the traits are controlled by QTL with large effects, among the established GWAS methods, weighted single-step GWAS is more suitable for the association study of domestic animals. This method has been applied to the growth, carcass and reproductive traits of livestock [11,12,13,14,15,16]. In addition, the candidate genes related to the QTL region identified in GWAS can be analyzed by gene networks. Gene networks are used to study the pathways and biological processes shared by these genes [17].

In this study, we used WssGWAS to detect genetic regions and further candidate genes related to semen traits in Duroc boars. This study will have an in-depth understanding of the genetic structure of semen traits and the biological information provided by gene networks and can be applied to speed up the genetic progress of semen traits in boars.

## 2. Materials and Methods

### 2.1. Population and Phenotype Data

Animals used in this study were from the same artificial insemination station (Guangzhou, China). A total of 24,983 semen records were collected from 583 Duroc pigs during 2020–2022. All boars had complete pedigree records for four generations. Four semen traits were measured. Sperm motility (SPMOT), sperm progressive motility (SPPMOT) and sperm abnormality rate (SPABR) were measured using Magapor Gesipor3.0 CASA system (Magapor S.L. Parque Científico Agroalimentario Valdeferrín-AulaDei, Ejea de los Caballeros, Zaragoza, Spain). Total sperm count (SPCOUNT) was calculated by multiplying semen volume (mL) by semen concentration (10^6^/mL, measured by the CASA system).

The phenotypes for four semen traits of Duroc pigs are shown in Table 1. According to the research of Marques et al. [18] and Wang et al. [19] and combined with the characteristics of our data, the quality control of phenotypes was: (1) frequency of semen collection <5 were excluded; (2) eliminating the semen records with an ejaculation volume ≤50 mL; (3) semen records with sperm motility <10% were excluded; (4) eliminating the semen records with adjacent semen collection interval >60 days and semen collection interval of 0 days.

### 2.2. Genotypic Data

Genomics DNA of pig semen samples were extracted and purified from 571 Duroc pigs. These boars were genotyped by using the KPS 50k SNP array (KPS Porcine Breeding Chip v1, Beijing, China) containing 57,566 SNPs. SNPs that unmapped the reference genome (*Sus scrofa* 11.1), and located in sexual chromosomes and with missing position information were removed, after which 50,897 SNPs were retained. After that, individuals with call rate <0.9, SNPs with call rate <0.9, minor allele frequency <0.01, and Hardy–Weinberg equilibrium <10^6^ were removed using plink v1.90 [20]. Finally, 38,054 SNPs were retained. Missing genotypes were finally imputed using Beagle software (version 4.1) [21].

### 2.3. Statistical analyses

Variance components and heritability of SPMOT, SPPMOT, SPABR, and SPCOUNT traits were estimated with two methods using the average information restricted maximum likelihood (AIREML) [22] of the AIREMLF90 procedure by BLUPF90 software [23]. The two methods are pedigree-based best linear unbiased prediction (BLUP) and ssGBLUP, and calculated genetic and phenotype correlations of semen traits using “OPTION se_covar_function.”

The following multiple traits repeatability model was used to estimate variance components:Y = Xβ + Za + Wp + Age + Intv + ε,(1)
where y is the vector of phenotypic observed value, β is the vector of fixed effects (year-season of semen collection and birth parities of boars), year is 2020–2022, March–May is spring, June–August is summer, September–November is autumn, December–February is winter, the birth parity of boars is 1–6, a~N(0, U$\sigma_{a}^{2}$) is a vector of additive genetic effects, U is pedigree-derived relationship matrix (A matrix) or H matrix, BLUP using A matrix and ssGBLUP using H matrix estimated variance components, p~N(0, I$\sigma_{p}^{2}$) is a vector of random permanent environmental effects, covariates Age and Intv denote the month age of the boars when semen collection and the interval between two subsequent semen collections in days, respectively, ε~N(0, I$\sigma_{e}^{2}$) is the vector of random residuals, and X, Z, and W are the incidence matrices of β, a, and p, respectively. The $\sigma_{a}^{2}$, $\sigma_{p}^{2}$ and $\sigma_{e}^{2}$ components are the additive genetic, permanent environmental and residual variances, respectively. I is the identity matrix. H is the matrix that combines pedigree and genomic information [10,24], and was calculated as:(2)$$H^{-1}=A^{-1}+\left[\begin{array}{cc} 0 & 0\\ 0 & G_{w}^{-1}-A_{22}^{-1} \end{array}\right]$$
where $A_{22}$ is a submatrix of A for the genotyped individuals: $G_{w}=0.9G+0.1A_{22}$. These weights were used to be compatible with genomic and phenotypic information and to control bias. $G=\frac{ZDZ^{\prime }}{\sum_{i-1}^{n}2p_{i}\left(1-p_{i}\right)}$ is the genomic relationship matrix [25], where Z is a genotype matrix adjusted for allele frequencies (with 0, 1, and 2 representing genotypes AA, Aa, and aa, respectively). D is a diagonal matrix containing the SNP weight, n is the number of SNPs, and $p_{i}$ is the minor allele frequency of the $i^{th}$ SNP.

The weighted single-step GWAS was conducted using the BLUPF90 software family in an iterative way adapted for genomic analyses [11]. Briefly, for phenotype, pedigree and genomic file preprocessing, we made use of RENUMF90 and variance components were estimated using AIREMLF90, which were then used in BLUPF90 to predict GEBV. SNP effects were then calculated using the postGSF90 [26] procedure. The association study was used for the iteration procedure according to Wang et al. [9] with the following steps:

Step 1: Procedure initialization, let *t* = 1, $D_{\left(t\right)}\;$= I, $G_{\left(t\right)}$ = λZ$D_{\left(t\right)}Z^{\prime }$ and λ =$\;\frac{1}{\sum_{i=1}^{n}2p_{i}\left(1-p_{i}\right)}$;

Step 2: Calculated GEBV of the entire dataset, via ssGBLUP method with $H^{-1}=A^{-1}+\left[\begin{array}{cc} 0 & 0\\ 0 & {\left(0.9G_{\left(t\right)}+0.1A_{22}\right)}^{-1}-A_{22}^{-1} \end{array}\right]$;

Step 3: Calculated marker effects, via ${\hat{g}}_{t}\;$= λ$D_{\left(t\right)}Z^{\prime }G_{\left(t\right)}^{-1}\hat{a}$, where $\hat{a}$ is the GEBV of the genotyped individuals;

Step 4: Calculated SNP weights for the next iteration, via $d_{i\left(t+1\right)}\;$=${\hat{\;g}}_{i\left(t\right)}^{2}2p_{i}\left(1-p_{i}\right)$;

Step 5: Normalized SNP weights, readjust the SNP weights to stabilize the total genetic variance via $D_{\left(t+1\right)}=\frac{tr\left(D_{\left(1\right)}\right)}{tr\left(D_{\left(t+1\right)}\right)}D_{\left(t+1\right)}$;

Step6: Calculated G for the next iteration, via $G_{\left(t+1\right)}=\lambda ZD_{\left(t+1\right)}Z^{\prime }$;

Step7: Let *t* = *t* + 1 and iterate from step 2.

This procedure was run for three iterations based on the predicted accuracies of GEBV according to Legarra et al. [27] and Zhang et al. [28], and was used by Wang et al. [9] and Marques et al. [5]. The SNP weights, G matrices, GEBV, and marker effects were updated at each iteration. Marker effects obtained from the third iteration were used to calculate the proportion of genetic variance explained by subsets of consecutive SNPs. The SNP window consisted of a region of consecutive SNPs located within 0.4 Mb, which is the average haplotype block size in commercial pig lines’ mated SNP effects [29,30] of Duroc pigs in this study. The genetic variance explained by the $i^{th}$ set of consecutive SNPs ($i^{th}$ SNP window) was calculated via:(3)$$\frac{Var\left(a_{i}\right)}{\sigma_{a}^{2}}\times 100\%=\frac{Var\left(\sum_{j=1}^{m}Z_{j}g_{j}\right)}{\sigma_{a}^{2}}\times 100\%$$
where $a_{i}$ is the genetic variance of the $i^{th}$ SNP window, $\sigma_{a}^{2}$ is the total additive genetic variance, $Z_{j}$ is the vector of the $j^{th}$ SNP for all individuals and $g_{j}$ is the effect of the $j^{th}$ SNP within the $i^{th}$ window. Manhattan plots of these windows were shown using the R software.

### 2.4. Candidate Gene Detection and Functional Enrichment Analysis

QTL regions were selected according to the genetic variance of chromosome windows. Windows explaining more than 1% genetic variance were selected as candidate QTL regions, within which candidate genes were searched. The threshold of 1% was chosen based on the literature [14,31,32] and the expected contribution of SNP windows. The expected proportion of average genetic variance explained by each window was 0.02% for the pigs (100/5039). The first three windows for single semen traits that explained the largest number of genetic variances were further extended to 0.4 Mb flanking regions of the midpoints both upstream and downstream.

Genome annotations were based on the gene database *Sus scrofa* 11.1 (http://www.ensembl.org, accessed on 11 October 2022). For all the candidate genes, we manually searched they National Center for Biotechnology Information (NCBI, http://www.ncbi.nlm.nih.gov, accessed on 11 October 2022) to see if they had a previously identified relationship with the traits under study. Gene Ontology (GO) [33] and Kyoto Encyclopedia of Genes and Genomes (KEGG) [34] were used for functional enrichment analysis of candidate genes.

## 3. Results

### 3.1. Descriptive Statistics and Genetic Parameters for the Semen Traits

Descriptive statistics of phenotypes for all semen traits are given in Table 1. The coefficients of variation (CV) for SPMOT, SPPMOT, SPABR, and SPCOUNT traits were 15.61%, 68.01%, 43.72%, and 53.36%, respectively. Among the semen traits, the coefficient of variation in sperm motility was the smallest. Semen records per animal min were 5, maximum 103, and mean ± SD 43.9 ± 26.2. A distribution histogram with ejaculation times is shown in Appendix A.

In order to better understand the genetic structure of semen traits, we used two methods (BLUP and ssGBLUP) to estimate variance components, the latter to calculate genetic and phenotypic correlation. The variance components of semen traits are shown in Table 2, and the genetic and phenotypic correlations are shown in Appendix A. Among all semen traits, the heritability estimated by ssGBLUP was lower than that estimated by BLUP, and the repeatability of the two methods almost the same. Among them, the heritability and repeatability of SPABR were the highest. There was a very strong negative genetic correlation between SPMOT and SPABR traits: −0.7351.

### 3.2. WssGWAS Results of Semen Traits

In this study, we show the proportion of variances explained by each 0.4 Mb window for semen traits of Duroc pigs (Figure 1). The three most important QTL regions and the candidate genes are shown in Table 3. The three most important windows of SPMOT, SPPMOT, SPABR and SPCOUNT explained 12.45%, 9.77%, 15.80%, and 12.15% of the genetic variance of each trait, respectively (Figure 1 and Table 3).

A total of 112 candidate genes were detected in the QTL regions of SPMOT, SPPMOT, SPABR and SPCOUNT traits (Appendix A), of which 9 genes were reported to be associated with mammal spermiogenesis, testes functioning, and male fertility (Table 3). Furthermore, 16, 17, 12, and 13 QTL regions (windows that explained more than 1% of total genetic variances) were found for SPMOT, SPPMOT, SPABR and SPCOUNT (Appendix A).

### 3.3. GO Terms and KEGG Pathway Enrichment Analysis

The GO terms used to enrich identified genes, with a total of four GO terms related to semen traits, are shown in Appendix A. The reproductive process, reproduction, meiotic cell cycle, and microtubule-based movement are important processes of the male reproductive process (Figure 2).

### 3.4. Association Network Diagram between GO Terms

The association network diagram of the GO terms of genes’ biological processes can be seen in Figure 3. The biological processes of the reproductive process cilium assembly, reproduction, and meiotic cell cycle work together to affect male meiotic nuclear division. In the mammal, the product of meiotic division of the male germ cell is the spermatozoon, and the union between the male and female germ cells in the process called fertilization results in the formation of a new organism.

## 4. Discussion

In this study, we chose the WssGWAS method because of the following advantages. (1) It can integrate all phenotypes, genotypes and pedigree data at the same time, thus avoiding the calculation of the pseudo-phenotypes of genotyped animals to integrate all phenotypic information. It uses information from those without genotypes to improve the statistical power of QTL detection. (2) It allows different weights to be used according to the importance of SNPs, which deviates from the unrealistic assumptions of the GBLUP infinitesimal model and improves the accuracy of SNP effect estimation [9]. (3) It offers the possibility of utilizing SNP windows, the percentage of genetic variance explained by a series of successive SNPs. Continuous SNP windows in GWAS may be more successful in finding QTL regions than single SNP analyses, due to linkage disequilibrium (LD).

The results of variance components showed that the repeatability of ssGBLUP was similar to that of BLUP and the additive variance of ssGBLUP lower than BLUP. Previous studies have shown that the additive variances and heritabilities estimated by the pedigree-based BLUP method may be too high. Compared with BLUP, ssGBLUP has a lower standard deviation [35]. In this study, we also found that the standard error estimated by ssGBLUP was smaller. That the ssGBLUP method uses both pedigree and genotype information to estimate the genetic parameters renders it more accurate in theory. The results of the two methods show that semen traits belong to medium heritability traits and great genetic progress can be obtained through selection. The heritabilities of SPMOT, SPPMOT, SPABR and SPCOUNT traits estimated by ssGBLUP were 0.168, 0.119, 0.244 and 0.177, respectively. This is basically consistent with the results of Gao et al. [6]: 0.160, 0.161, 0.261 and 0.183.

In this study, we identified several QTL regions related to semen traits in Duroc pigs. The search regions of candidate genes are not only limited to the SNP window but also include upstream and downstream flanking regions. It is important to use larger genomic regions to identify genes, because the SNPs in the window may be in the high LD and the QTL in the surrounding regions. We found the same QTL region (chr18:14.23–14.60) in SPMOT and SPPMOT, and determined that the *STRA8* gene is a candidate gene for these two traits. The same QTL region (chr12:58.83–59.21) was found in SPMOT and SPABR, and the candidate genes of *ZSWIM7*, *TEKT3* and *UBB* were identified, which may be related to the high genetic correlation between SPMOT and SPABR (Appendix A). These results confirm the hypothesis of gene pleiotropy.

The *STRA8*, *UBB* and *CATSPER1* genes are enriched into a common biological process and participate in male meiotic spermatogenesis. In order to further confirm the identified candidate genes related to semen traits, further molecular experiments are needed in future research.

For sperm motility, *STRA8*, *ZSWIM7*, *TEKT3*, *UBB*, and *CATSPER1* are significant candidate genes. *STRA8* (stimulated by retinoic acid 8) controls the initiation of meiosis in male germ cells by activating the expression of meiotic genes [36]. The spermatozoa of *STRA8* knockout mice can eventually form sperm cells that cannot complete meiosis [37], which will increase germ cell apoptosis. This is known as the goalkeeper of male meiosis [38]. It has been confirmed that it is an important candidate gene affecting spermatogenesis in mice [39,40,41] and Atlantic salmon [42]. Recurrent *ZSWIM7* (zinc finger SWIM-type containing 7) mutations lead to human male infertility [43]. The homozygous variation of *ZSWIM7* leads to azoospermia in men and primary ovarian insufficiency in women [44]. *TEKT3* (tektin 3) is a filamentous protein associated with microtubules in cilia, flagella, basal bodies and centrioles. *TEKT3* is necessary for progressive sperm motility in mice [45]. The translocation of *TEKT3* in bull spermatozoa may be related to capacitation or overactivation [46]. The expression of the polyubiquitin gene *UBB* plays an important role in maintaining RNA binding regulatory factors and piRNA-metabolic proteins in testis to complete mouse spermatogenesis [47]. The targeted destruction of the polyubiquitin gene *UBB* leads to male and female infertility in mice, and germ cells are blocked during meiotic prophase I [48]. The *CATSPER1* (cation channel, sperm associated 1) protein is localized in sperm tail and expressed in human testicular tissue in the form of meiosis and postmeiosis [49]. There is high expression of *CATSPER1* in human azoospermic semen [50]. The downregulation of *CATSPER1* channel in epididymal sperm contributes to the pathogenesis of asthenospermia in rats [51,52,53].

For sperm progressive motility, *PTBP2* and *STRA8* are significant candidate genes. The splicing regulation of *PTBP2* (polypyrimidine tract binding protein 2) is very important for the communication between germ cells and Sertoli cells (multifunctional somatic cells necessary for spermatogenesis) [54]. The RNA binding protein *PTBP2* plays an important role in the development of male germ cells and spermatogenesis in mice [55,56].

For sperm abnormality rate, *EIF2B2*, *MLH3*, *ZSWIM7*, *TEKT3*, and *UBB* are significant candidate genes. In human male asthenospermia, *EIF2B2* (eukaryotic translation initiation factor 2B subunit beta) is important for sperm motility [57,58,59]. The loss of function of the DNA mismatch repair gene *MLH3* can lead to male infertility with azoospermia or severe oligozoospermia.

For total sperm count, *CCDC70* is a significant candidate gene. *CCDC70* (coiled-coil domain containing 70) gene is highly expressed in mouse testis, mainly expressed in sperm cells, round sperm group and epididymal epithelial cells, and participates in the regulation of spermatogenesis and epididymal sperm maturation.

## 5. Conclusions

We used weighted single-step GWAS to identify candidate genes related to sperm motility, sperm progressive motility, sperm abnormality rate, and total sperm count in Duroc boars. These results can be used as genetic markers to improve semen production and quality. This study provides information for further understanding the genetic structure of semen traits in Duroc boars.

## Figures and Tables

**Figure 1 animals-13-00365-f001:**
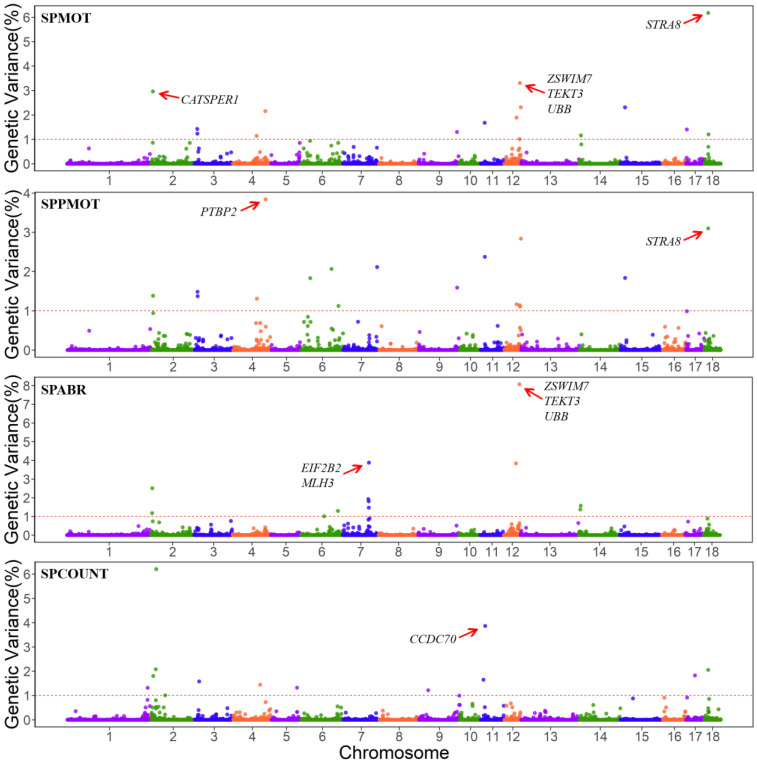
Manhattan plot for GWAS results of semen traits in Duroc pigs.

**Figure 2 animals-13-00365-f002:**
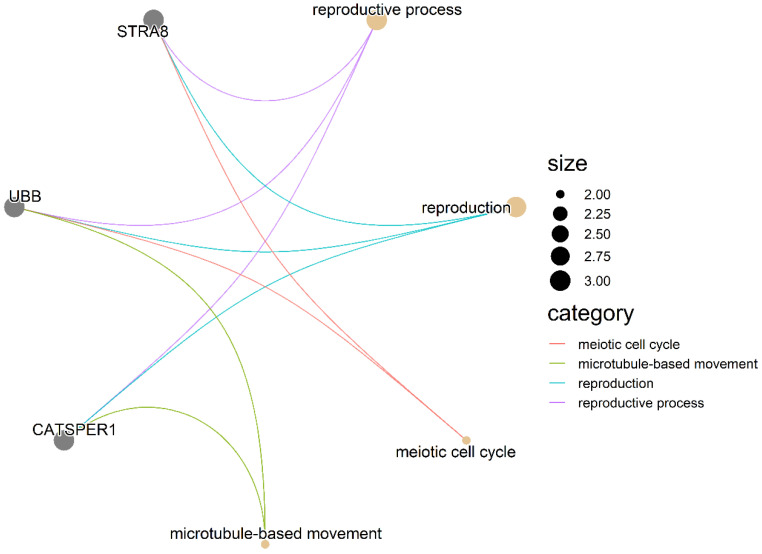
Gene network of biological processes for semen traits of candidate genes. Gray nodes represent candidate genes, yellow nodes represent biological processes, and different biological processes and candidate genes are connected by different colors.

**Figure 3 animals-13-00365-f003:**
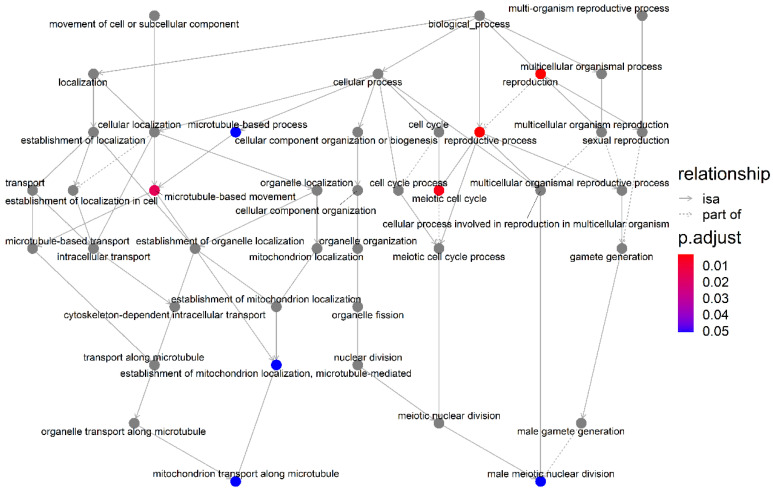
Association network diagram between GO terms of enrichment genes. The solid line represents the complete correlation of the two biological processes, and the dotted line represents the partial correlation. The color represents the size of the adjusted *p* value.

**Table 1 animals-13-00365-t001:** Descriptive statistics for the semen traits of Duroc pigs.

Trait	Number of Boars	Number of Phenotypes	Mean ± SD	CV(%) ^a^	Minimum	Maximum
SPMOT/%	583	24,833	82.35 ± 12.08	14.67	10.00	100.00
SPPMOT/%	583	24,730	28.84 ± 18.44	63.92	1.00	100.00
SPABR/%	583	24,917	28.81 ± 12.49	43.35	4.00	100.00
SPCOUNT/10^8^	583	24,492	364.67 ± 190.59	52.26	50.00	2529.09

^a^ CV: Coefficient of variation.

**Table 2 animals-13-00365-t002:** Genetic parameters for the semen traits of Duroc pigs.

Trait	Models	$\sigma_{a}^{2}$(SE)	$\sigma_{p}^{2}$(SE)	$\sigma_{e}^{2}$(SE)	$h^{2}$(SE)	$r_{e}$(SE)
SPMOT	BLUP	50.472 ± 11.526	15.672 ± 7.563	110.890 ± 1.026	0.285 ± 0.058	0.374 ± 0.020
	ssGBLUP	29.428 ± 6.584	34.956 ± 4.471	110.860 ± 1.025	0.168 ± 0.034	0.367 ± 0.018
SPPMOT	BLUP	67.082 ± 14.463	7.460 ± 9.143	258.340 ± 2.394	0.202 ± 0.040	0.224 ± 0.016
	ssGBLUP	39.267 ± 8.315	31.831 ± 5.095	258.310 ± 2.394	0.119 ± 0.023	0.216 ± 0.014
SPABR	BLUP	65.056 ± 17.395	47.887 ± 12.085	71.246 ± 0.658	0.353 ± 0.082	0.613 ± 0.019
	ssGBLUP	44.741 ± 11.239	67.419 ± 7.871	71.246 ± 0.658	0.244 ± 0.054	0.612 ± 0.017
SPCOUNT	BLUP	9436.900 ± 2092.000	2706.900 ± 1358.500	22884.000 ± 213.240	0.269 ± 0.053	0.347 ± 0.020
	ssGBLUP	6116.500 ± 1247.900	5610.700 ± 782.780	22881.000 ± 213.200	0.177 ± 0.032	0.339 ± 0.018

$\sigma_{a}^{2}$, Genetic variance; $\sigma_{p}^{2}$, variance of environmental effect; $\sigma_{e}^{2}$, residual variance; $h^{2}$, heritability; $r_{e}$, Repeatability; SE, standard error.

**Table 3 animals-13-00365-t003:** Three most important QTL regions and candidate genes for semen traits of Duroc pigs.

Traits ^a^	Chr ^b^	Position (Mb)	gVar (%) ^c^	Nsnp	Candidate Genes
SOMOT	18	14.23–14.60	6.18	13	*STRA8*
	12	58.83–59.21	3.31	10	*ZSWIM7, TEKT3, UBB*
	2	6.19–6.58	3.87	6	*CATSPER1*
SPPMOT	4	121.17–121.57	3.84	10	*PTBP2*
	18	14.23–14.60	3.10	13	*STRA8*
	12	62.16–62.44	2.84	6	*-*
SPABR	12	58.83–59.21	8.07	10	*ZSWIM7, TEKT3, UBB*
	7	97.93–98.26	3.88	10	*EIF2B2, MLH3*
	12	46.05–46.45	3.84	7	*-*
SPCOUNT	2	17.69–18.09	6.21	8	*-*
	11	15.98–16.37	3.86	10	*CCDC70*
	2	15.98–16.09	2.08	3	-

^a^ SPMOT: sperm motility; SPPMOT: sperm progressive motility; SPABR: sperm abnormality rate; SPCOUNT: total sperm count. Within each trait, genomic regions were decreasingly sorted based on the proportion of genetic variance explained; ^b^ Chr: chromosome; ^c^ gVar (%): proportion of genetic variance explained by 0.4 Mb.

## Data Availability

The datasets generated and/or analyzed during the current study are not publicly available, since the studied population consists of the nucleus herd of Guangdong Guyue Technology Co., Ltd., China, but are available from the corresponding author on reasonable request.

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
