# Peer review of "Identification of New Candidate Genes Related to Semen Traits in Duroc Pigs through Weighted Single-Step GWAS"

_animals, 2023, doi:10.3390/ani13030365_

Round 1

Reviewer 1 Report

Zheng et al. conducted an association study for several key semen traits in Duroc pics by using weighted single-step GWAS. They identified several candidate genes, many of which appear to have functions related to spermatogenesis and meiosis. The estimated heritability for the four traits is largely consistent with previous reports, confirming the validity of their methodological approaches.

The manuscript is clearly written, and the conclusions are adequate. I do not have many suggestions to offer. One thing that I would like to see is if their results can be used for developing a polygenic prediction model for selecting better-performing semen for artificial insemination.   

Specific comments:

Line 122-123: Where is the relationship matrix term in the equation, corresponding to U?

Author Response

Thanks for your careful review and constructive suggestions. Please see the attachment for a detailed reply.

Reviewer 2 Report

The introduction should be improved by giving more attention to the justification of the importance of the characteristics under study and what is the state of knowledge on the subject. The methodology of analysis can be referred to in the section on materials and methods.

It would be appropriate to discuss further why the variance components are larger with one method than the other.

The discussion of results is good, however, some of the conclusions seem to be a recounting of results and some are implications of the results.

Author Response

(The authors gave the same response as above.)

Reviewer 3 Report

The main objective of manuscript entitled “Identification of new candidate genes related to semen trait in Duroc pigs through Weighted single-step GWAS” is detection of single nucleotide polymorphisms (SNPs) associated with semen traits in Duroc pigs. Generally, pig breeding companies consider Duroc as a paternal line in their breeding strategies. Therefore, improvement of semen traits such as sperm motility, abnormality rate of sperm, and total sperm count could be important for breeders due to their effects on success in artificial insemination. The manuscript is informative and could be helpful for pig breeders. However, there are some minor and major concerns which are summarized below:

Major:

1.      Please provide the minimum, average and maximum number of records per animal. Also, provide number of levels for fixed effects and depth of pedigree.

2.      As your results are shown in Table 2, BLUP model could explain a higher range of heritabilities than that of ssGBLUP. I wonder what if you use deregressed EBVs/EBVs as input and then run a simple GWAS analysis. When BLUP model could explain a higher range of genetic variation than that of using ssGBLUP, could we expect to have more reliable results using a simple GWAS (pedigree-based EBVs/deregressed EBVs consider as input in a simple GWAS) than WssGWAS or not?

 Minor:

1.      Please change “semen trait” to “semen traits” in the title of your manuscript.

2.      Please change it (in line 46): “artificial pig breeding”

Author Response

(The authors gave the same response as above.)

Round 2

Reviewer 3 Report

The authors applied most of my concerns. However, there is still one of my major concerns:

"When BLUP model could explain a higher range of genetic variation than that of using ssGBLUP, could we expect to have more reliable results using a simple GWAS (pedigree-based EBVs/deregressed EBVs consider as input in a simple GWAS) than WssGWAS or not?"

The authors answered this question: “the GWAS results using DEBV derived by BLUP have higher type I error.”

I am not sure if this answer could be comprehensive. Thus, authors must comprehensively provide evidence (references), and discuss how GWAS results using DEBV derived by BLUP have higher type I error.

Author Response

Thank you for your comments and suggestions.We have replied to all your comments. Please refer to the attachment for detailed reply.

Round 3

Reviewer 3 Report

To me, the authors' answers were convincing. But, readers may have the same questions. Therefore, authors must apply their answers in the manuscript.